# Targeting Diabetic Retinopathy with Human iPSC-Derived Vascular Reparative Cells in a Type 2 Diabetes Model

**DOI:** 10.3390/cells14171352

**Published:** 2025-08-30

**Authors:** Sergio Li Calzi, Dibyendu Chakraborty, Ping Hu, Ram Prasad, Yvonne Adu-Rutledge, Cristiano Vieira, Fadeela Sheini, Michael E. Boulton, Mervin C. Yoder, Changde Cheng, Maria B. Grant

**Affiliations:** 1Department of Ophthalmology and Visual Sciences, University of Alabama at Birmingham (UAB), Birmingham, AL 35294, USA; scalzi@uabmc.edu (S.L.C.); dchakrab1@gmail.com (D.C.); huping3366@gmail.com (P.H.); ramprasad@uabmc.edu (R.P.); yadu-rutledge@meei.harvard.edu (Y.A.-R.); cristianobio@gmail.com (C.V.); fsheini@uab.edu (F.S.); meboulton@uabmc.edu (M.E.B.); 2Schepens Eye Research Institute, Harvard Medical School, and Massachusetts General Hospital, Boston, MA 02114, USA; 3Department of Surgery, McGowan Institute for Regenerative Medicine, University of Pittsburgh, Pittsburgh, PA 15213, USA; myoder@iu.edu; 4Department of Surgery, Indiana Center for Regenerative Medicine and Engineering, Indiana University School of Medicine, Indianapolis, IN 46202, USA; 5Department of Hematology & Oncology, University of Alabama at Birmingham (UAB), Birmingham, AL 35294, USA; ccheng@uabmc.edu

**Keywords:** KNA cells, CD34 cells, endothelial colony forming cells, inducible pluripotent stem cells, diabetic retinopathy, vascular repair

## Abstract

**Purpose:** To investigate the therapeutic potential of inducible pluripotent stem cell (hiPSC)-based vascular repair, we evaluated two vascular reparative cell populations, CD34^+^ cells derived from hiPSC (hiPSC-CD34^+^) and endothelial colony forming cells (ECFCs) derived from hiPSC (iPS-ECFCs), alone and in combination, in a type 2 diabetic (db/db) mouse model of DR. **Methods:** hiPSC-CD34^+^ cells (1 × 10^4^) or iPSC- ECFCs (1 × 10^5^) alone or in combination (1.1 × 10^5^) were injected into the vitreous of immunosuppressed db/db mice with six months of established diabetes. One month post-injection, mice underwent electroretinography (ERG) and optical coherence tomography (OCT) to evaluate functional and structural retinal recovery with iPSC administration. Immunohistochemistry (IHC) was used to assess recruitment and incorporation of cells into the retinal vasculature. Retinas from the experimental groups were analyzed using Functional Proteomics via Reverse Phase Protein Array (RPPA). **Results:** Functional assessment via ERG demonstrated significant improvements in retinal response in the diabetic cohorts treated with either hiPSC-derived CD34^+^ cells or hiPSC-ECFCs. Retinal thickness, assessed by OCT, was restored to near-nondiabetic levels in mice treated with hiPSC-CD34^+^ cells alone and the combination group, whereas hiPSC-ECFCs alone did not significantly affect retinal thickness. One month following intravitreal injection, hiPSC-CD34^+^ cells were localized to perivascular regions, whereas hiPSC-ECFCs were observed to integrate directly into the retinal vasculature. RPPA analysis revealed interaction-significant changes, and this was interpreted as a combination-specific, non-additive host responses (m^6^A, PI3K–AKT–mTOR, glycolysis, endothelial junction pathways). **Conclusions:** The studies support that injection of hiPSC-CD34^+^ cells and hiPSC-ECFCs, both individually and in combination, showed benefit; however, iPSC combination-specific effects were identified by measurement of retinal thickness and by RPPA.

## 1. Introduction

Diabetic retinopathy (DR), the most common diabetic microvascular complication [1], is the leading cause of acquired vision loss worldwide in middle-aged and therefore economically active individuals. DR accounts for 4.8% of the number of cases of blindness (37 million) worldwide [2,3,4], with the number of individuals with vision-threatening DR estimated to rise to 191 million by 2030 [5]. Importantly, the presence of DR indicates microcirculatory dysfunction in other organ systems [6]. Despite better control of the modifiable risk factors, such as glucose, blood pressure, and lipids, and improved screening programs, DR remains a global health issue. Current therapies for these retinal conditions include laser photocoagulation, injection of anti-VEGF antibodies, or vitreoretinal surgery; though partially effective, these therapies carry significant side effects and do not correct the underlying retinal pathology. To date, cell therapy to rescue blood flow to the ischemic retina has involved autologous CD34^+^ cells [7,8,9]. Alternative and more effective autologous cell therapies are thus urgently needed, using cell populations that can withstand a diseased retinal environment and generate robust and functional vessels repairing degenerate capillaries. Furthermore, DR and microvascular complications are less amenable to pharmacotherapy due to the irreversible nature of diabetes and the long-term epigenetic changes, and thus may be more amenable to cell therapy, including iPSC approaches, that could result in partial shedding of epigenetic changes during reprogramming of these cells [10].

Vascular repair involves coordinated action between hematopoietic cells (such as myeloid angiogenic cells (MACs) or CD34^+^ cells) and vascular wall-derived stem/progenitor cells (endothelial colony forming cells, ECFCs). In vivo, MACs, a subset of hematopoietic cells, support vascular repair by releasing factors that stimulate resident endothelial cell proliferation. In humans, MACs are typically characterized by CD34^+^ expression. When isolated from individuals with microvascular complications, CD34^+^ cells are dysfunctional, demonstrating reduced migration, altered paracrine function, and inability to correct capillary vasodegeneration in murine models of retinal disease [11]. In both human subjects and murine models, ECFCs reside within the endothelial lining of vessels (from capillaries to conduits) to participate in endothelial repair or replacement, but can be mobilized as circulating ECFCs entering the circulation and migrating into areas of vascular injury [12,13]. ECFCs are severely depleted in diabetics with microvascular complications [14] and, unlike ECFCs from control subjects, cannot be cultured from the peripheral blood [11]. Human hiPSC-ECFCs display the ability to form functional and durable blood vessels in vivo and, importantly, confer therapeutic revascularization by connecting with and remaining integrated with host vessels long term [15]. Previously, we showed that hiPSCs can be differentiated into a mesodermal subset marked by kinase insert domain receptor (KDR), neural cell adhesion molecule (CD56), and apelin receptor (APLNR), and called KNA^+^ cells. KNA^+^ cells give rise to ECFCs in vivo and to both endothelial cells and smooth muscle cells (SMC)/pericytes in vitro [16]. KNA^+^ cells exhibited high clonal proliferative potential and differentiated into ECFCs. When implanted subcutaneously into immunodeficient mice, they form perfused blood vessels and successfully repair retinal vasculature when injected into the vitreous of type 2 diabetic (db/db) mice, restoring perfusion and correcting vascular dysfunction.

Previously, we tested the combination of freshly isolated CD34^+^ cells and cultured ECFCs in the murine model of retinopathy of prematurity and demonstrated marked improvement in pups treated with this combination of donor cells, showing reductions in retinal oxidative stress and inflammation [17]. Park et al. demonstrated long-term in vivo engraftment of hiPSC-derived vascular progenitors into the retina. The hiPSC-derived vascular progenitors were generated using viral-integrated and non-integrated fibroblast as well as cord blood-derived hiPSC lines. These cells successfully homed to and engrafted in the murine retina in an ischemia-reperfusion (I/R) injury model [7,18]. In the current study, we evaluated the efficacy of the combination of hiPSC-CD34^+^ cells and hiPSC-ECFCs alone and in combination on structural and functional changes in the diabetic retina using a murine model of type 2 diabetes.

## 2. Materials and Methods

### 2.1. hiPSC Generation and Culture

CD34^+^ cells and ECFCs were generated from hiPSCs following previously published protocols [19]. Briefly, human peripheral blood mononuclear cells (PBMCs) from control donors were stored in liquid nitrogen and shipped to ALSTEM, Inc. (Richmond, CA, USA) for reprogramming into hiPSCs and for characterization. iPSCs were maintained in mTeSR^TM^ Plus Media on Matrigel purchased from STEMCELL Technologies (Seattle, WA, USA) in a 37 °C and 5% CO_2_ environment. Cell culture media were replaced daily, and cultures were examined for the elimination of differentiated cells. Ten days later, large colonies showed an embryoid body (EB) in their core. Seven of the EBs were collected and seeded on a new dish coated with gelatin in mTeSR^TM^ Plus Media and regular media (1:1 ratio) overnight. Directed differentiation of human pluripotent stem cells (hPSC) into the mesoderm (Stage-Specific Embryonic Antigen-5 (SSEA)-5^−^KNA^+^) and ECFC (NRP-1^+^CD31^+^) lineages was performed as previously detailed in Gil et al. [16] CD34^+^ cells were differentiated using the protocol of Piau et al. [20] and enriched using EasySep™ Human CD34 Positive Selection Kit II (Stem cell technologies, Cambridge, MA, USA). Human samples, peripheral blood, were obtained from diabetic and age- and gender-matched healthy individuals under the Institutional Review Board-approved protocol (IRB-300000173) at the University of Alabama at Birmingham (UAB).

### 2.2. Animal Study

Ten-week-old male db/db (BKS.Cg-Dock7m +/+ Leprdb/J) Strain #000642 and control db/m mice were obtained from Jackson Laboratories (Bar Harbor, ME, USA) and maintained in a 12 h light/12 h dark, temperature, and humidity-controlled environment throughout the study. Blood glucose was evaluated weekly, and the mice were considered suitable as a model of diabetes if glycemia was above 13.88 mmol/L on two consecutive days. Animal experimental protocol (#20919) was approved by the Institutional Animal Care and Use Committee (IACUC) at the University of Alabama at Birmingham (UAB). All mice were immunosuppressed with cyclosporine A (30 mg/kg) on alternate days by intraperitoneal injection, starting one week before intravitreal injection and continuing for an additional week post-injection. hiPSC-CD34^+^ cells (1 × 10^4^) and hiPSC-ECFCs (1 × 10^5^) were injected alone or in combination (1.1 × 10^5^) in a 1 µL saline solution into the vitreous of db/db mice with 6 6-month duration of diabetes. In selected experiments, KNA^+^ cells were injected at a concentration of 1 × 10^5^ and all controls were saline-injected. The concentration of cells was based on previous studies performed using freshly isolated peripheral blood-derived CD34^+^ cells and culture-derived ECFCs [17]. The number of injected db/db mice per group was as follows: Saline: *n* = 7; iPSC-KNA^+^: *n* = 8; iPSC-CD34^+^: *n* = 5; iPSC-ECFCs: *n* = 4; iPSC-CD34^+^/iPSC-ECFCs: *n* = 5. Injected db/m mice: Saline: *n* = 5.

### 2.3. Electroretinogram (ERG)

ERGs were performed using a LKC Bigshot ERG system. Briefly, mice were dark-adapted overnight. The animals were anesthetized with ketamine (80 mg/kg total body weight, BW) and xylazine (15 mg/kg total BW), then pupils were dilated with atropine/phenylephrine under dim red light. Once dilated, animals were exposed to 5 full-field white light flashes at 0.25 and 2.5 cd·s/m^2^ under scotopic conditions. The animals were then light-adapted for 5 min and exposed to 10–15 full-field white light flashes at 10 and 25 cd·s/m^2^ under photopic conditions. Responses were averaged and analyzed using the LKC EM software (LKC Technologies, Gaithersburg, MD, USA).

### 2.4. Immunohistochemistry

Mice were euthanized, and eyes were immediately enucleated. The globes were punctured behind the limbus with a 30G needle to allow the fixative to reach the posterior chamber. Eyes were then fixed in 4% paraformaldehyde (PFA) buffered solution for 4 h at 4 °C, transferred to Phosphate Buffered Saline (PBS) solution, and the neuroretinas dissected. After a brief rinse in PBS, retinas were transferred to a permeabilization/blocking solution (0.1% Triton x-100 in PBS and either 10% normal Donkey or Goat serum) (Cat. # D9663, G9023, Sigma-Aldrich, St. Louis, MO, USA) for 3 h at 4 °C. Samples were then incubated overnight at 4 °C in primary antibodies: Mouse anti-human Nuclear Antigen (HNA) (dil. 1:50) (Cat. # ab-191181, Abcam, Waltham, MA, USA), rabbit anti-Collagen IV (dil. 1:100) (Cat # ab-19808, Abcam), and rabbit anti-NG2 (dil. 1:100) (Cat. # AB5320, Sigma-Aldrich). After three washes in PBS, samples were incubated in secondary antibodies: Goat anti-rabbit IgG (H+L) AlexaFluor™ 488 (dil. 1:500) (Cat.# A-21206, ThermoFisher Scientific Inc.) and Donkey anti-mouse IgG (H+L) AlexaFluor™ 555 (dil. 1:500) (Cat. # A-31570, ThermoFisher Scientific Inc., Waltham, MA, USA) for one hour at room temperature. After another cycle of PBS washes, retinas were coverslip-mounted using Vectashield^®^ Antifade mounting medium with DAPI (4′, 6-diamidino-2-phenylindole) for nuclear staining (Cat. # H-1200, VectorLabs, Newark, CA, USA). Images were captured using the Zen image acquisition software on a Zeiss Axio Imager Z2 (Zeiss, Oberkochen, Germany) epifluorescence microscope.

Cell integration analysis was performed using Fiji (ImageJ 1.54p). RGB images were first converted to a 16-bit binary format. The green channel, representing blood vessels, was thresholded to quantify the vascular area, which was normalized to 100%. The red channel, corresponding to integrated and perivascular cells, was similarly thresholded to determine the total cell area. Perivascular cells were manually traced to calculate their specific area, and the area of integrated cells was derived by subtracting the perivascular cell area from the total cell area.

### 2.5. Optical Coherence Tomography (OCT)

OCT imaging was performed as described previously [21]. Briefly, eyes were dilated with a 1:1 mixture of 1% tropicamide and 2.5% phenylephrine hydrochloride. Mice were anesthetized with 100 mg/kg ketamine and 5 mg/kg xylazine. 2.5% hydroxypropyl methylcellulose (Gonak; Akorn, Lake Forest, IL, USA) was applied to each eye. OCT images were obtained using a Bioptigen OCT system (Envisu R-class; Leica Microsystems, Wetzlar, Germany). A lateral image, averaged across 100 B-scans, was obtained at the centermost position of the optic nerve head. OCT images were imported to ImageJ 1.53j and the images were binarized. The portion containing the choroid was retained, and the thickness of the retina to the left and right of the optic nerve was measured at each pixel along the *x*-axis. The resulting average of these measurements was used as the average retinal thickness for each eye.

### 2.6. Proteomic Analysis

For protein expression assays, the retinas from the experimental cohorts were disrupted in the TissueLyser LT (QIAGEN) in 200 μL of lysis buffer provided by the Functional Proteomics Reverse Phase Protein Array (RPPA) Core Facility (MD Anderson Cancer Center): 1% Triton X-100, 50 mM Hepes (pH 7.4), 150 mM NaCl, 1.5 mM MgCl_2_, 1 mM EGTA, 100 mM NaF, 10 mM Na pyrophosphate, 1 mM Na_3_VO_4_, 10% glycerol, and freshly added protease and phosphatase inhibitors (Roche Diagnostics Corporation, Indianapolis, IN, USA). After pelleting the debris, the protein concentration was determined with the Pierce BCA microplate procedure (Thermo Fisher Scientific, Waltham, MA, USA) on a Synergy H1 plate reader (BioTek Instruments, Inc., Winooski, VT, USA), adjusted to approximately 1 μg/μL and sent to RPPA Core for proteomic analysis using their standard procedures (https://www.mdanderson.org/research/research-resources/core-facilities/functional-proteomics-rppa-core.html). Data were analyzed as described for gene expression using Ingenuity Pathway Analysis (QIAGEN, Germantown, MD, USA) [17]. Heatmaps were created in R using the ggplot2 package per standard protocols.

### 2.7. Statistical Analysis

All experiments were performed in triplicate. Results are expressed as means ±SEM. The power of analysis with a 95% confidence interval was used to calculate the sample size required to obtain statistically significant results. The sampling number we used gave a normal distribution. Significance of differences was assessed by a two-tailed Student’s *t* test or one-way ANOVA–Tukey post hoc test, multiple comparison test, or chi-square test. Statistical analysis was done by calculating the *p*-values (α = 0.05) using GraphPad Prism 9 version 9.2.0 (332) software. Data determined to be normally distributed was assessed for statistical significance by a two-tailed unpaired parametric *t* test. Data determined to be non-normally distributed was assessed for statistical significance by a two-tailed Mann-Whitney nonparametric unpaired *t*-test. RPPA data were analyzed using a two-factor linear model with terms for CD34 status, ECFC status, and their interaction; *p*-values were adjusted for multiple testing, and proteins were declared significant at adjusted *p* < 0.01 and |log2FC| ≥ 1.1.

## 3. Results

### 3.1. KNA^+^ Cells Differentiate into Pericytes in db/db Mice with DR

Previously, we utilized hiPSCs and differentiated them into a specific mesoderm subset characterized by KNA^+^ expression. We demonstrated that by day 4 in culture, iPSCs differentiated into KNA^+^ cells were able to generate both endothelial cells and pericytes/smooth muscle cells when embedded in collagen gel and implanted subcutaneously in NOD-SCID mice [16]. In this study, we asked if day 4 KNA^+^ cells could also differentiate into pericytes within the retina. To address this, we injected 1 × 10^5^ KNA^+^ cells intravitreally into diabetic (db/db) and control (db/m) mice. Saline-injected db/m and db/db mice served as controls, and mice were euthanized at 1 or 2 months post-intravitreal injection. As shown in Figure 1A–C, KNA^+^ cell injections into the vitreous of db/db mice with 6 months of diabetes showed individual cells and clumps of perivascular cells that co-stain for human nuclear antigen (HNA). In db/m mice, the cells remained on the retinal surface, resembling cellular sheets, and were not associated with the vasculature (Figure 1D). This behavior is not unexpected, as in the db/m mice, no retinal vascular injury exists, and thus, the KNA^+^ cells have no signal to migrate into the retina to repair (Figure 1D). In contrast, HNA^+^ cells (Figure 1E–J) stain for the pericyte-specific marker NG-2, indicating the human origin of pericytes in the db/db mouse retina.

### 3.2. hiPSC-CD34^+^ and hiPS-ECFCsAlone and in Combination Resulted in Correction of Diabetes Induced Structural and Functional Retinal Pathology

In Figure 2A–C, as expected, saline injections in db/db mice showed no HNA^+^ cells within the retina. One month post-injection, the combination of ECFCs and CD34^+^ cells showed evidence of vessel repair (yellow arrowheads) and the presence of perivascular cells (blue arrowheads) (Figure 2D–F). As shown (Figure 2G–I), hiPSC-ECFCs formed robust vasculature by 2 months post-injection. In Figure 2J–L, hiPSC-CD34^+^ cells were detected in perivascular locations within the retinas of db/db mice up to 2 months post-injection, supporting the long-term survival of this population. Quantification of cellular integration in retinas of mice injected intravitreally with iPSC-CD34^+^ and iPSC-ECFC demonstrated that the majority of injected iPSCs representing ECFCs integrated into the vasculature, while a smaller proportion representing CD34^+^ cells remained in a perivascular location (integrated: 27.8 ± 15.5%; Perivascular: 3.4 ± 1.5%; *n* = 3) (Appendix A).

To assess whether hiPSCs injections resulted in functional improvement of the retina, mice underwent ERGs and OCT testing. One month after administering the hiPSC-CD34^+^, a significant improvement was observed in the scotopic a-wave (189.5 μV ± 20.92 vs. 296.2 μV ± 26.15, *p* = 0.024, *n* = 7) (Figure 3A,C) and b-wave (261.7 μV ± 32.55, *n* = 7 vs. 472.7 μV ± 33, *p* = 0.0003, *n* = 8) (Figure 3A,D). hiPSC-ECFCs administration (261.7 μV ± 32.55, *n* = 7 vs. 493.7 μV ± 20.55, *p* = 0.0001, *n* = 5) and the combination of hiPSC-CD34^+^ and hiPSC-ECFCs (261.7 μV ± 32.55, *n* = 7 vs. 353.0 μV ± 40.43, *p* = 0.05, *n* = 9) significantly improved the scotopic b-wave compared to saline-injected db/db mice (Figure 3A,D). A significant improvement was also observed in the photopic b-wave in the hiPSC-CD34^+^ injected mice (89.8 μV ± 9.87, *n* = 8 vs. 114.7 μV ± 7.28, *p* = 0.03, *n* = 8) (Figure 3B,E). To assess preservation of the structure of the retina, OCT imaging was performed (Figure 3F–N). In chronic diabetes, thinning of the inner retinal layer, inner nuclear layer, photoreceptor layer, and outer plexiform layer is commonly observed [22]. When compared to saline-injected mice, total retinal thickness significantly improved in mice injected with hiPSC-CD34^+^ (202.02 μm ± 0.73, *n* = 8 vs. 207.6 μm ± 1.26, *n* = 7, *p* = 0.0003), ECFCs (202.02 μm ± 0.73, *n* = 5 vs. 207.45 μm ± 1.90, *n* = 9, *p* = 0.013), while the combination treatment preserved retinal thickness even closer to nondiabetic controls (Figure 3N).

### 3.3. Proteomic Analysis of Signaling Mechanisms in iPSC-Injected db/db Retinas

The unexpected observation that CD34^+^ cells alone provided more functional improvement, as assessed by ERG, than the combination of CD34^+^ cells and ECFCs prompted us to investigate differences in signaling mechanisms. To this end, we performed functional proteomic analysis using RPPA on retinal tissues from the five experimental cohorts. Differential expression analysis, clustering, and pathway enrichment were conducted to interpret the biological significance of the observed changes. Principal component analysis (Figure 4A) reveals treatment-driven separation along PC1 (35.1% of total variance), with saline, iPSC-CD34^+^ cells, and iPSC-ECFC samples clustering together on the left side while combination treatment samples occupy a distinct position on the right side. PC2 (24.95% of variance) captures primarily intra-group heterogeneity rather than inter-treatment differences. As shown in Figure 4A, the complexity of cellular signaling increased progressively from saline-treated controls to hiPSC-CD34^+^ cells, hiPSC-ECFCs, and, as expected, was highest in the combination group.

A total of 293 features were associated with the transition from baseline to the combination treatment (Appendix A). Among the top signaling proteins identified were: (i) 3-phosphoinositide-dependent protein kinase 1 (PDPK1), a serine/threonine protein kinase that plays a crucial role in the PI3K/Akt signaling pathway, insulin signaling and vascular patterning; (ii) Programmed cell death ligand 1 (PD-L1) which is an immune-suppressive protein that protects against endothelial dysfunction and modulates T-cell behavior; (iii) Mitf, a transcription factor that affects development of the retinal vasculature, is associated with vascular abnormalities in mice carrying pathogenic variants in this gene [23] and Mitf can regulate blood retinal barrier (BRB) function [24]; (iv) Catenin β-1 (CTNNB1) modulates growth and development of the blood vessels in the retina [25]; (v) Rho GTPase Activating Protein 45 (ARHGAP45) is a Rac-GAP that contributes to the balance between formation and disruption of endothelial tight and adherens junctions; (vi) AlkB Homolog 5, RNA Demethylase(ALKBH5) is associated with retinal pigment epithelium anomalies and choroidal neovascularization in age-related macular degeneration and dysregulation of the AKT/mTOR pathway and (vii) Synaptophsin is an integral membrane protein of small synaptic vesicles and can track the formation of synapses in the outer plexiform layer and inner plexiform layer.

Exogenous hiPSCs administration activates Ras, Rap1, insulin, neurotrophins, ErbB, Fox, HIF-1, and phospholipase D (Figure 4B). These signaling molecules are part of, or interact with, the PI3K-Akt, MAPK/ERK, and the mTOR pathways that are central to retinal cell proliferation, survival, metabolism, and vasculogenesis. Insulin, neurotrophins, and ErbB receptors, such as epidermal growth factor receptors, activate downstream signaling cascades involving Ras, Rap1, and PI3K/Akt. These cascades influence transcription factors, including FoxO and HIF-1, which regulate gene expression in response to growth signals and hypoxia. Expectedly, the hiPSC treatments activated the angiogenesis and hypoxia response in the retina. HIF-1 is a master regulator of the cellular response to low oxygen and promotes angiogenesis. Phospholipase D can modulate mTOR signaling and participates in membrane dynamics and cell migration, which are important for vessel formation. Ras and Rap1 are small GTPases that function as molecular switches in many of these signaling pathways, including those triggered by insulin and growth factors. FoxO transcription factors are downstream targets of Akt and engage in oxidative stress resistance, metabolism, and apoptosis. Collectively, these proteins and signaling cascades participate in fundamental retinal processes ranging from development and cell survival to immune regulation, vascular integrity, and synaptic communication.

### 3.4. Protein Expression Profiling Reveals Interactional Effects of Combination CD34^+^ and ECFC Therapy

Of 420 antibodies/proteins quantified, combination therapy significantly altered 45 proteins meeting the fold-change threshold (|log2FC| > 1.0, adjusted *p* < 0.01), including 42 downregulated and 3 upregulated proteins. Neither monotherapy condition produced detectable changes at these thresholds in whole retinal tissue, suggesting cell-type-specific effects were diluted below detection in bulk tissue analysis. The effects observed only with combination treatment indicate that the co-occurrence of both cell types (iPSC-ECFCS and IPSC-CD34^+^ cells drives detectable host tissue remodeling. The widespread downregulation likely reflects resolution of diabetic stress responses and metabolic normalization through combination therapy (Figure 5).

Metabolic normalization represented a major therapeutic mechanism involved in the benefits observed with the iPSC cell combination, with coordinated suppression of glycolytic enzymes typically elevated in diabetic retinopathy, such as Hexokinase 2 (HK2), Pyruvate Kinase M2 (PKM2), Lactate Dehydrogenase A (LDHA), and Phosphoribosyl aminoimidazole Carboxylase/Synthetase (PAICS). The amino acid transporter SLC1A5, often upregulated under metabolic stress, similarly decreased. These enzymes have distinct transcriptional regulation in photoreceptors and under diabetic conditions. HK2 is regulated by AKT signaling, with phosphorylated AKT promoting its mitochondrial translocation to inhibit apoptosis; its transcription is suppressed by high glucose via PPARγ, reducing its protective role in diabetes [26]. PKM2, the predominant isoform in photoreceptors, is downregulated in diabetic retinopathy, impairing rod function; it also acts as a transcriptional regulator in its dimeric form, linking metabolism to gene expression [27]. LDHA supports aerobic glycolysis in photoreceptors and facilitates NAD^+^ regeneration; its disruption affects cone morphology, highlighting metabolic interdependence [28]. PAICS, involved in purine biosynthesis, lacks detailed transcriptional data in the retina but may be indirectly regulated by metabolic stress and transcription factors like CRX [29]. In diabetic retinas, chronic hypoxia and nutrient stress drive pathological upregulation of glycolysis. The coordinated suppression observed here indicates improved tissue oxygenation and restored oxidative metabolism, representing therapeutic benefit.

Vascular-related proteins showed the largest magnitude changes in the dataset, indicating substantial vascular remodeling. VASP phosphorylation at S239, associated with vascular permeability in diabetes, decreased dramatically (log2FC = −10.12, adjusted *p* = 1.58 × 10^−6^). This represents the largest change observed and likely reflects restoration of endothelial barrier integrity. Supporting vascular normalization, β-catenin (CTNNB1) was among the few significantly upregulated proteins (log2FC = 2.15, adjusted *p* = 8.49 × 10^−4^). Complementing this, ARHGAP45, a negative regulator of vascular stability through Rho GTPase signaling, showed substantial suppression. CTNNB1 (β-catenin) modulation influenced retinal blood vessel growth and development [30], while ARHGAP45 contributed to endothelial junction balance [31]. Together, these changes indicate a coordinated program of vascular stabilization.

The m6A RNA methylation machinery showed significant coordinated regulation. ALKBH5, an m6A demethylase, showed the strongest suppression among all proteins analyzed (log2FC = −4.73, adjusted *p* = 8.73 × 10^−6^). Additional m6A pathway components were coordinately downregulated: WTAP, METTL3, FTO, and YTHDF3. This coordinated suppression of both writers (WTAP) and erasers (ALKBH5, FTO) alongside readers (YTHDF3) suggests reduced m6A turnover, potentially stabilizing specific transcripts involved in therapeutic response.

Multiple stress response pathways showed coordinated downregulation, consistent with tissue recovery. DNA damage markers decreased significantly: RAD23A, phospho-H2AX, and ATM. Heat shock protein signaling similarly decreased, with phospho-HSP27 showing a substantial reduction. Synaptophysin (SYP), which tracks synapse formation in retinal plexiform layers [32], showed suppression (log2FC = −4.07, adjusted *p* = 2.17 × 10^−4^). Given the functional improvements observed in parallel studies, this likely represents synaptic reorganization rather than loss. In diabetic retinopathy, synaptic proteins are often pathologically elevated due to compensatory mechanisms. The observed decrease may indicate normalization toward healthy baseline levels.

Growth factor and survival signaling showed comprehensive modulation. MAPK/ERK signaling pathway components showed coordinated regulation through RAF1 as a key kinase in the cascade [33]. MYC and ERBB3 all decreased, suggesting reduced proliferative signaling as tissue stabilizes. The mTOR pathway showed coordinated suppression, including TSC2 and phospho-4E-BP1, indicating metabolic quiescence rather than active growth signaling. Anti-apoptotic proteins, including BCL2, showed decreased expression alongside cell cycle regulators such as PLK1 [34], suggesting reduced cellular turnover and proliferation as the tissue stabilizes rather than active pro-survival signaling.

## 4. Discussion

The findings of this study highlight that the diabetic retinal environment is permissive for vascular integration of iPSC-derived vascular reparative cells. iPSC-CD34^+^ cells alone provide significant functional benefit as assessed by ERG, potentially via paracrine signaling. Combination therapies enhanced retinal structure as assessed by IHC and OCT. Proteomic analysis identified key pathways involved in vascular remodeling and immune modulation, and the interplay of these complex pathways is critical for maintaining retinal structure/function.

Previously, we showed that healthy peripheral blood-derived (nondiabetic) CD34^+^ cells home to areas of injury in the diabetic retina and in the I/R retinal injury model [35]. CD34^+^ cells represent a population of progenitor cells that are hematopoietic in origin and express CD45^+^ and VEGF receptors, among other surface receptors. They consistently contribute to revascularization by providing paracrine support to the resident vasculature; however, in chronic diseases such as diabetes, they are dysfunctional and no longer facilitate repair.

ECFCs represent a second [14,35,36,37] population of cells possessing vascular reparative activity [38,39]. ECFCs, also called “outgrowth endothelial cells”, Ref. [39] have high proliferative capacity and the ability to maintain an endothelial phenotype throughout ex vivo long-term expansion [40]. These cells are capable of integrating into pre-existing retinal vessels, de novo retinal capillary formation in several in vitro models, and de novo capillary formation in in vivo retinal models [41,42,43]. Furthermore, ECFCs injected into the systemic circulation of SCID mice are able to correct ischemia and lodge and survive in nine different vascular beds for up to 7 months after intravenous tail vein injection, without inducing thrombosis or infarcts [44]. Following acute vascular injury, the ability of ECFCs to repair and regenerate can be augmented by the addition of healthy CD34^+^ cells [45]. Previously, we showed that ECFCs could only be obtained from diabetic individuals who do not have microvascular complications, whereas ECFCs could be obtained from 100% of controls [11]. Thus, the use of iPSC-CD34^+^ and iPS-ECFCsis needed for vascular repair in diabetics with complications, as their endogenous cells are severely dysfunctional and unable to repair injured cells.

Our previous studies showed that KNA^+^ cells can generate robust ECFCs that can restore retinal perfusion and correct vascular dysfunction in diabetic mice [16]. In this study, we show that KNA^+^ cells represent a population that can replace pericytes in the diabetic retina. Pericytes are contractile cells that wrap around the endothelial cells of capillaries and venules throughout the body, including the retina. Pericytes are essential for stabilizing blood vessels, regulating the BRB [46], controlling capillary blood flow, and supporting endothelial survival and function [47]. One of the earliest detectable changes in DR is loss of pericytes, which leads to the classic histological features of DR, including capillary instability, microaneurysm formation, and increased vascular permeability [48].

The signaling studies performed provide an unbiased assessment of signal transduction and cellular regulation in the retina. Collectively, these results suggest that treatment with hiPSC CD34^+^ cells and hiPSC-ECFCs in the DR model of db/db mice results in beneficial effects on retinal neurons and synaptic plasticity, as Ras, Rap1, and neurotrophins are central to these processes. The exogenous administration of these cells also improves metabolic regulation as demonstrated by activation of insulin and FoxO, and enhances cell proliferation (e.g., Ras, ErbB, HIF-1) to promote cell repair or replacement. While DR has been considered a microvascular disease of the retina, evidence supports that retinal neurodegeneration is an early and significant event in its progression, often preceding visible blood vessel abnormalities [49]. The combination of iPSC-CD34^+^ cells and iPSC-ECFCs produced a synergistic therapeutic effect not observed with either cell type alone, indicating a statistically significant interaction driven by intercellular crosstalk. This joint treatment activated distinct molecular pathways including RNA methylation (via ALKBH5, FTO, YTHDF3, WTAP), growth factor signaling (ERBB3, IGF1R, mTOR/PI3K/Akt), metabolic reprogramming (HK2, PKM2, LDHA, PAICS), vascular remodeling (CTNNB1, ARHGAP45), neuronal enhancement (SYP), MAPK/ERK signaling (RAF1, DUSP6, SRC), and cell survival mechanisms (MCL1, BCL2, PLK1) highlighting a coordinated and multifaceted response unique to the combination therapy.

Thus, diabetes directly affects the health and function of the retinal neurons, leading to early vision changes and contributing to irreversible vision loss, particularly through damage to retinal ganglion cells and their axons. Administration of hiPSC-CD34^+^ cells resulted in ERG improvement. An increase in the scotopic a-wave primarily reflects changes in the activity of rod photoreceptors, while the enhancement in the photopic b-wave supports cone ON-bipolar cells. While ON-bipolar cells are the main contributors of the photopic b-wave, Müller glial cells are also thought to play a significant role in shaping the b-wave, as the Müller cells respond to extracellular potassium ions released by depolarizing bipolar cells [50]. hiPSC-ECFCs administration improved scotopic-b waves, suggesting that hiPSC-ECFCs beneficially impact rod ON-bipolar cells, the primary cells that receive input from rod photoreceptors in the dark-adapted retina. While rod ON-bipolar cells are the direct source of the scotopic b-wave signal, Müller glial cells also play a significant role in shaping and amplifying the b-wave. Müller cells function as “spatial buffers” for potassium ions. When bipolar cells depolarize, they release potassium into the extracellular space. Müller cells absorb this potassium and redistribute it, creating a current flow that contributes to the b-wave [50].

hiPSCs have great utility as a cell therapy due to their inexhaustible self-renewal and proliferation ability [51]. To date, their use for the possible treatment of ophthalmic diseases represents a significant portion of hiPSC clinical trials [52]. iPSCs have been differentiated into mesoderm and subsequently into retinal vascular endothelial using combinations of VEGF, FGF, Norrin, and vitronectin for repair of the vasculature [53]. Strategies to target neurons include the generation of iPSC-derived retinal ganglion cells [54]. “Disease-in-a-Dish” models use patient-derived iPSCs and represent a unique opportunity to enhance cell replacement but also show benefit for drug testing [55]. iPSC-based approaches for large production and banking of specific epithelial, glial, or neuronal retinal cells are now available to model diseases, including retinal pathology such as diabetic retinopathy. iPSC-derived retinal cell libraries offer new biological tools to not only understand different retinal diseases but also to develop new treatments adapted to each specific disease.

This study has several limitations, including the use of only male mice and the short duration of the study, representing only two months of follow-up on the impact of the cells. When quantifying intravascular vs. perivascular cells (Appendix A), we could not distinguish iPSC-ECFCs that had not yet integrated from iPSC-CD34^+^ cells, as both cells were labeled with HNA. Finally, the outcome in the diabetic retinopathy model might have been different if we had used iPSCs derived from a diabetic donor rather than the healthy control donor iPSCs used in this study.

## 5. Conclusions

In conclusion, iPSC vascular reparative cells provide marked benefit to the neural retina. Early KNA^+^ cells can not only generate ECFCs, but also pericytes that provide structural support to the vessels undergoing vasodegeneration. The combination of hiPSC-CD34^+^ cells and hiPSC-ECFCs had the most dramatic impact on the preservation of retina thickness. Collectively, this work supports the feasibility of using hiPSC-CD34^+^ cells and hiPSC-ECFCs differentiated from allogeneic iPSCs for a future treatment strategy to combat the complex pathophysiology associated with DR.

## Figures and Tables

**Figure 1 cells-14-01352-f001:**
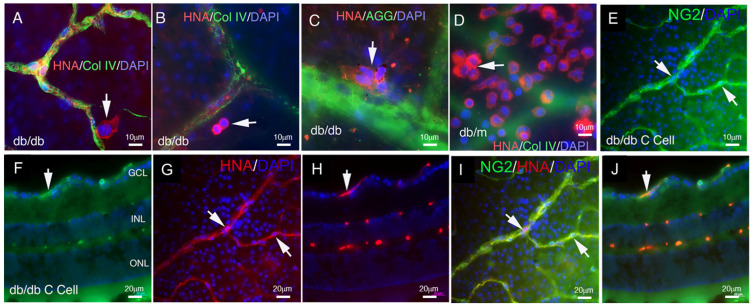
**KNA^+^ cells migrate to areas of retinal injury in db/db mice and become pericytes.** Representative confocal images of retinal flat mounts from db/db mice injected with KNA^+^ cells from healthy donors. KNA^+^ cells migrate deep into the retina of db/db mice and accumulate next to the damaged blood vessels (**A**–**C**). Conversely, in db/m (non-diabetic mice), the cells remain on the retinal surface (**D**). Retinas are stained with collagen IV (**A**,**B**,**D**) or agglutinin (AGG) (**C**) for the detection of blood vessels (green), and with human nuclear antigen (HNA, red) to identify the iPSC-derived KNA^+^ cells of human origin. Confocal images of retinal flat mounts (**E**,**G**,**I**) and cross sections (**F**,**H**,**J**) from db/db mice injected with KNA^+^ iPSC. The retinas are stained with NG2 for the detection of pericytes (green), and with human nuclear antigen (HNA, red) to identify cells of human origin. Arrows highlight human cells. Nuclei were stained with DAPI (blue). The number of injected mice in each group was as follows: db/db iPSC-KNA^+^: *n* = 8; db/m iPSC-KNA^+^: *n* = 8.

**Figure 2 cells-14-01352-f002:**
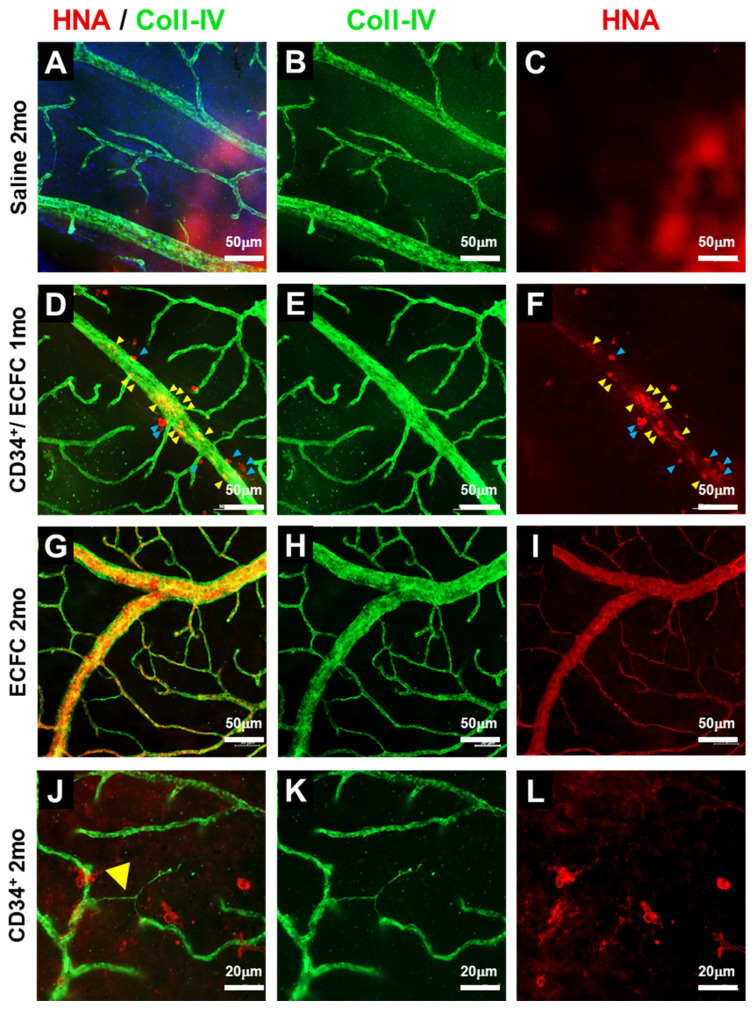
**One- and two-month time points after stem cell injections.** iPSC-CD34^+^ cells migrate to injured blood vessels while iPS-ECFCsincorporate into the damaged vessel walls of diabetic mice. Representative confocal images of retinal flat mounts from diabetic mice injected with Saline (**A**–**C**) or the combination of iPSC-ECFC and iPSC-CD34^+^ cells (**D**–**F**) or iPSC-ECFC (**G**–**I**) or iPSC-CD34^+^ cells (**J**–**L**). Retinas are stained with collagen-IV for the detection of blood vessels (green), and with HNA (red) to identify cells of human origin. Yellow arrowheads (**D**,**F**) highlight iPSC ECFCs incorporated in the blood vessels, while blue arrowheads (**D**,**F**) indicate iPSC-CD34^+^ cells with perivascular localization. Yellow arrowhead (**J**) indicates an acellular capillary, the hallmark of diabetic retinopathy. The number of injected db/db mice in each group was as follows: Saline: *n* = 7; iPSC-CD34^+^: *n* = 5; iPSC-ECFCs: *n* = 4; iPSC-CD34^+^/iPSC-ECFCs: *n* = 5.

**Figure 3 cells-14-01352-f003:**
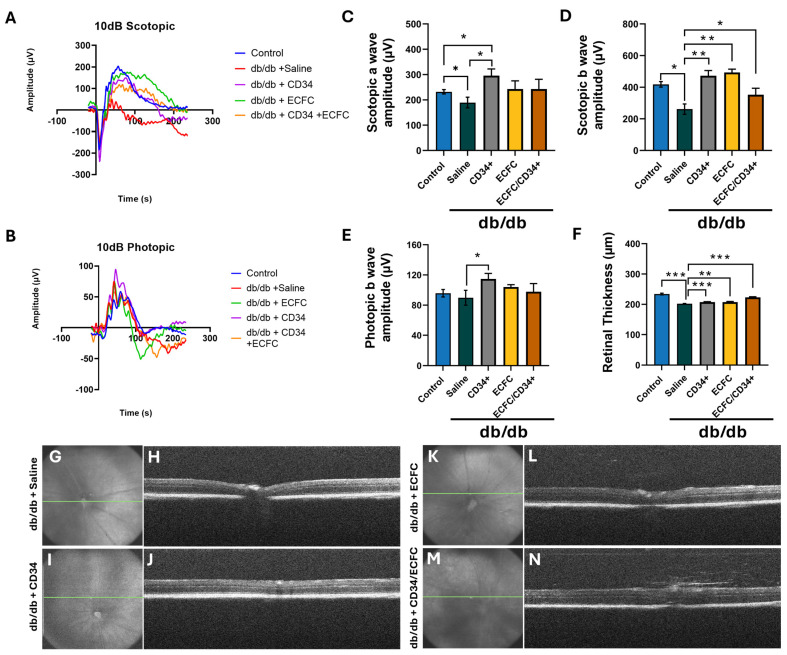
ERG and OCT changes in hiPSC-treated mice. Whole animal dark-adapted (**A**) and light-adapted (**B**) flash ERGs were performed on db/db mice. Mean waveforms in response to the 10dB light flash intensity evaluated and quantification of Scotopic a-wave amplitudes (**C**), Scotopic b-wave amplitudes (**D**), and Photopic b-wave amplitudes (**E**). Quantification of retinal thickness by OCT (**F**). En-face view of the posterior segment with the optic nerve head. The green line indicates the area of the retina where the B-scan image was taken (**G**,**I**,**K**,**M**). Representative OCT B-scan images of mouse retina from cross-sectional scan of en-face images from db/db mice injected with saline (**H**), iPSC-CD34^+^ cells (**J**), iPSC-ECFC (**L**), and the combination of iPSC-CD34^+^ cells and iPSC-ECFCs (**N**). Retinal thickness was assessed by OCT. When compared to saline-injected mice, total retinal thickness significantly improved in mice injected with hiPSC-CD34^+^, hiPSC-ECFCs, and the combination of cells. * *p* ≤ 0.05, ** *p* ≤ 0.005, and *** *p* ≤ 0.0005. The number of injected db/db mice in each group was as follows: Saline: *n* = 7; iPSC-CD34^+^: *n* = 5; iPSC-ECFCs: *n* = 4; iPSC-CD34^+^/iPSC-ECFCs: *n* = 5. Injected db/m mice: Saline: *n* = 5.

**Figure 4 cells-14-01352-f004:**
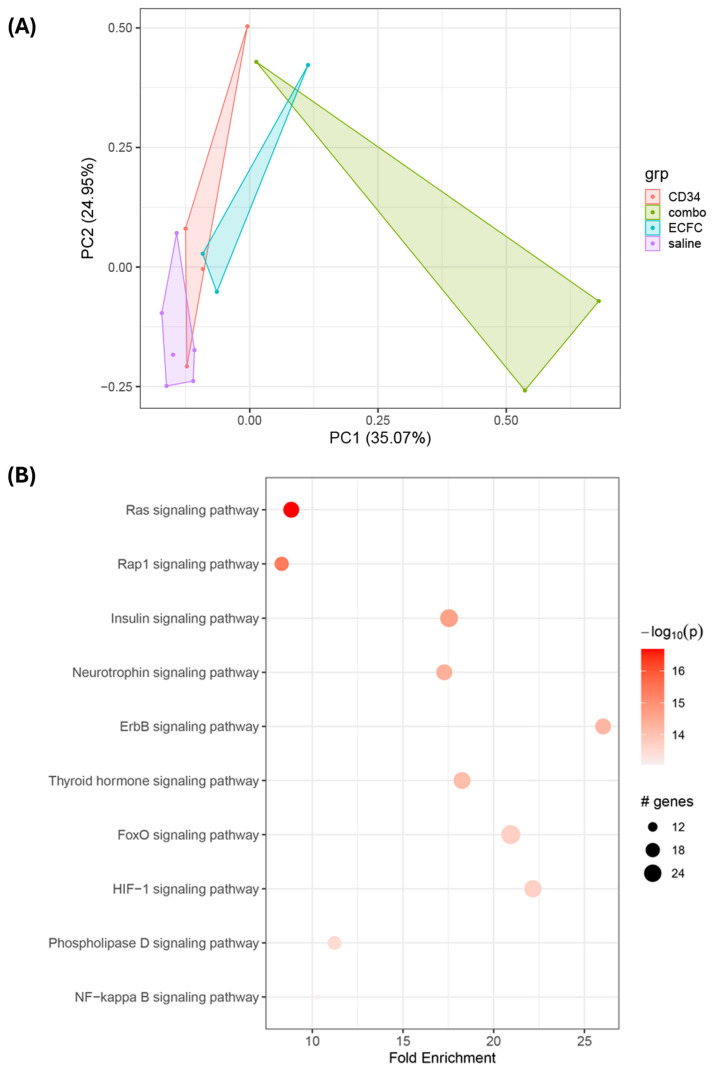
**Trajectory analysis using the RPPA data and Phase Protein Array.** Trajectory-Associated Pathway enrichment analysis (**A**) showing signaling pathways significantly associated with the pseudotime trajectory derived from principal curve analysis. The chart displays the top 10 signaling pathways (filtered for terms containing “signaling”) ranked by statistical significance. Protein expression changes along the trajectory were tested for association with canonical pathways using pathfindR. This identifies key biological processes that are dynamically regulated along the inferred treatment progression pathway. Principal Component Analysis (PCA) (**B**) of reverse-phase protein array (RPPA) data from 15 samples across four treatment groups: saline control (*n* = 5), iPSC-CD34^+^ cells (*n* = 4), iPSC-ECFC (*n* = 3), and combination treatment (combo, *n* = 3). Each point represents a sample, colored by treatment group. The plot shows the first two principal components, which capture the major sources of variation in the protein expression data. Polygon frames outline the boundaries for each group. The analysis reveals distinct clustering patterns among treatment groups, with a clear separation between the combination treatment and other conditions. Principal curve analysis was fitted to the first two principal components of the RPPA data. Blue points represent individual samples projected onto the principal curve (pink line, smoothed using smooth spline method with stretch parameter = 2). Dark red points show the projection of each sample onto the fitted curve. Sample labels indicate cell types. The principal curve represents a trajectory through the data that captures the major pattern of variation, potentially representing a treatment progression pathway. This analysis provides a pseudotime ordering of samples based on their position along the curve.

**Figure 5 cells-14-01352-f005:**
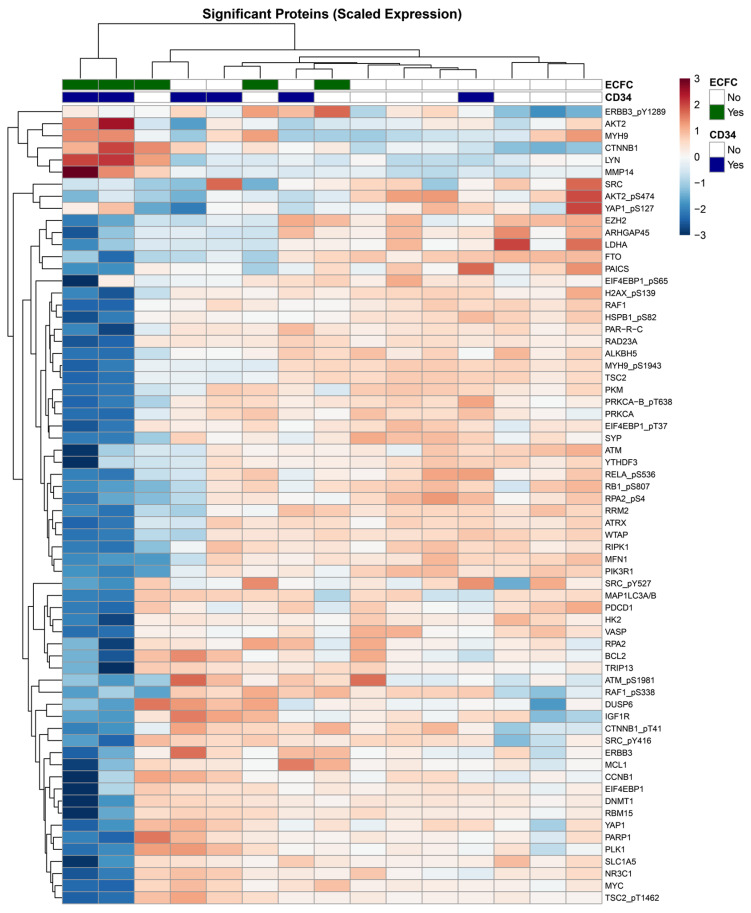
**Heatmap of gene expression profiles in iPSC-ECFC versus Combination treatment conditions.** Heatmap showing expression patterns of 70 significant protein forms across samples grouped by treatment conditions in a 2 × 2 experimental design (CD34^+^/CD34^−^ and ECFC^+^/ECFC^−^). Rows represent individual proteins (including total and phosphorylated forms), and columns represent individual samples. Expression values were z-score normalized across rows to show relative expression differences within each protein. The color scale reflects z-scores, with blue indicating below-average expression and red indicating above-average expression for each protein across all samples. Hierarchical clustering grouped proteins by similarity in expression patterns and organized samples based on treatment status.

## Data Availability

Data are available upon reasonable request from Dr. Grant.

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
