# Peer review of "Targeting Diabetic Retinopathy with Human iPSC-Derived Vascular Reparative Cells in a Type 2 Diabetes Model"

_cells, 2025, doi:10.3390/cells14171352_

Round 1
Reviewer 1 Report
Comments and Suggestions for Authors
This study presents a strong and timely contribution with potential for significant impact in the field. While the foundational data are compelling, several areas require refinement to strengthen the manuscript.
I respectfully recommend that the authors incorporate these changes to strengthen the manuscript’s clarity and ensure its suitability for publication.
-
Proteomic Analysis Sample Size: The proteomic experiments were conducted with only 2–3 biological replicates per group, which is insufficient for robust analysis in high-dimensional datasets. Please provide a rationale for this limited sample size, or consider expanding the number of replicates to enhance statistical power.
-
Unexpected ERG Outcome with CD34⁺ Cells: The observation that CD34⁺ cells alone produced superior ERG improvements compared to the combination therapy warrants further mechanistic insight. Could this be due to competition between cell types, spatial crowding within the retina, or immune-mediated effects?
-
Quantification of Cellular Integration: While the imaging suggests reparative activity, the study would be strengthened by quantifying perivascular localization of HNA⁺ cells and the extent of ECFC integration (e.g., the percentage of vessels associated with human cells).
-
Notation and Dosage Clarity: The reported cell dosages are inconsistently presented—e.g., CD34⁺ as 1×10⁴, ECFCs as 1×10⁵, and combination as 1.1×10⁵, versus occasional use of shorthand like “105.” Please ensure consistent and precise notation throughout the manuscript to avoid confusion.
-
Early Definition of KNA⁺ Cells: The term “KNA⁺ cells” should be defined clearly and earlier in the introduction. Consider introducing the defining markers (KDR⁺, CD56⁺, APLNR⁺) when first mentioned to aid reader understanding.
-
Image Quality and Labeling: Confocal images should be provided in higher resolution suitable for publication. Additionally, all figures must include scale bars for proper spatial reference.
-
Contextualization in the Literature: The discussion would benefit from a deeper comparison with existing iPSC-based therapeutic strategies for diabetic retinopathy, particularly those targeting neuronal as well as vascular components of the disease.
Author Response
- Proteomic Analysis Sample Size: The proteomic experiments were conducted with only 2–3 biological replicates per group, which is insufficient for robust analysis in high-dimensional datasets. Please provide a rationale for this limited sample size or consider expanding the number of replicates to enhance statistical power.
Response: We agree that proteomic studies benefit from larger numbers of biological replicates. In our case, the use of primary CD34⁺ cells and ECFCs constrained the feasible “n” per group; to increase statistical power under these constraints we adopted a 2×2 factorial design with four groups (control, CD34 alone, ECFC alone, CD34+ECFC). This design allows us to estimate the main effects of each factor and their interaction using all available samples (total N=15 across the study), rather than relying solely on pairwise contrasts within single groups (n=2–3).
Analytically, we fit a linear model to the RPPA intensities with terms for the two main effects and their interaction, and we control false discovery using multiple-testing adjustment. RPPA’s focused, targeted panel (~400 proteins) and low technical variance make it well-suited to such factorial modeling. To further guard against false positives with modest n, we pre-specified conservative reporting criteria (|log₂FC| ≥ 1 and adjusted p < 0.01). Under these criteria, we identify differentially expressed protein probes as follows: 45 (combination vs control), 35 (combination vs ECFC), and 45 (combination vs CD34). These results are consistent in direction across biological replicates.
We have clarified these design and analysis choices in the Methods and acknowledge the sample-size limitation in the Discussion.
We added one sentence to Methods: “RPPA data were analyzed using a two-factor linear model with terms for CD34 status, ECFC status, and their interaction; p-values were adjusted for multiple testing, and proteins were declared significant at adjusted p < 0.01 and |log₂FC| ≥ 1.”1 “
- Unexpected ERG Outcome with CD34⁺ Cells: The observation that CD34⁺ cells alone produced superior ERG improvements compared to the combination therapy warrants further mechanistic insight. Could this be due to competition between cell types, spatial crowding within the retina, or immune-mediated effects?
Response: The reviewer mentions some interesting possibilities regarding why the iPSC-CD34+ cells have superior ERG improvements. The reviewer’s suggestions of competition between cells, spatial crowding within the retina or immune mediated effects are certainly valid. Indeed, the mechanism of iPSC-CD34+ cells being the most beneficial for improving ERGs may also lie in that these cells produce trophic factors and secrete within the area that the migrate into- which includes the majority of the retinal cells -rod photoreceptors (the main contributors to the scotopic a-wave when stimulated by light and to a lesser extent the ON bipolar cells and Muller glial cells). Since the iPSC-CD34+ cells also impact the photopic b- wave they also impact cone photoreceptors, ON bipolar cells (cone type) and the Müller glial cells. While the iPCS-ECFCs only migrate to particular areas of vascular injury and revascularize these regions to increase perfusion, the impact of the iPSC-CD34 cells likely covers a greater area as these cells secrete many diffusible factors (both growth factors and anti-inflammatory cytokines), thus they may influence the photoreceptor layer improving the scotopic a-wave ERGs and also the photopic-b wave demonstrating the impact on cone-driven ON bipolar cells and Muller cells.
Quantification of Cellular Integration: While the imaging suggests reparative activity, the study would be strengthened by quantifying perivascular localization of HNA⁺ cells and the extent of ECFC integration (e.g., the percentage of vessels associated with human cells).
Response: We have added the quantification of the perivascular vs. integrated cells in the cohort of mice that received the combination of both iPCS-CD34+ cells and iPSC-ECFCs.
Methods: Cell integration analysis was performed using Fiji (ImageJ). RGB images were first converted to 16-bit binary format. The green channel, representing blood vessels, was thresholded to quantify the vascular area, which was normalized to 100%. The red channel, corresponding to integrated and perivascular cells, was similarly thresholded to determine the total cell area. Perivascular cells were manually traced to calculate their specific area, and the area of integrated cells was derived by subtracting the perivascular cell area from the total cell area. (lines 167-171)
Results: Supplemental Figure1 shows the image analysis of retinas of mice injected intravitreally with iPSC-CD34⁺ and iPSC-ECFC revealed that the majority of injected cells representing ECFCs integrated into the vasculature, while a smaller proportion representing CD34+cells remained in a perivascular location (Integrated: 27.8% ± 15.5; Perivascular: 3.4% ± 1.5; n = 3). (lines 215-219)
- Notation and Dosage Clarity: The reported cell dosages are inconsistently presented—e.g., CD34⁺ as 1×10⁴, ECFCs as 1×10⁵, and combination as 1.1×10⁵, versus occasional use of shorthand like “105.” Please ensure consistent and precise notation throughout the manuscript to avoid confusion.
Response: We appreciate the reviewer bringing this to our attention and we now have standardized all the doses of cells.
- Early Definition of KNA⁺ Cells: The term “KNA⁺ cells” should be defined clearly and earlier in the introduction. Consider introducing the defining markers (KDR⁺, CD56⁺, APLNR⁺) when first mentioned to aid reader understanding.
Response: We described KNA cells in the introduction (lines 88-94) in the original manuscript but now we have expanded on them also describing them in the results lines 211-213.
- Image Quality and Labeling: Confocal images should be provided in higher resolution suitable for publication. Additionally, all figures must include scale bars for proper spatial reference.
Response: We have improved the quality of the confocal images and include high magnification images. Now all the figures have scale bars.
- Contextualization in Literature: The discussion would benefit from a deeper comparison with existing iPSC-based therapeutic strategies for diabetic retinopathy, particularly those targeting neuronal as well as vascular components of the disease.
Response: We have added a paragraph line 406-414 that compares existing iPSC-based approaches for diabetic retinopathy.
Reviewer 2 Report
Comments and Suggestions for Authors
The manuscript presents a well-designed and clearly written study evaluating the therapeutic potential of human inducible pluripotent stem cell-derived CD34⁺ cells and endothelial colony-forming cells (ECFCs), both alone and in combination, for vascular repair in a murine model of diabetic retinopathy, and provides valuable insights into the mechanisms underlying retinal recovery, however, I suggest a few minor revisions that may enhance the clarity and impact of the work, for instance, it would be helpful to ensure consistent use and definition of abbreviations such as hiPSC, CD34⁺, ECFCs, KNA⁺, and DR at their first mention in the abstract and throughout the main text to facilitate understanding, in the methods section, consider clarifying the rationale for the chosen dosages and combination ratios of injected cell types, since this may interest readers wishing to replicate or compare protocols, additionally, although the proteomic analysis via RPPA is described in detail, a brief explanation of how pathway enrichment and clustering were performed and interpreted would make this section more accessible to non-specialist readers, for the results, the text could benefit from more explicit cross-referencing to the corresponding figures and tables, especially when discussing quantitative improvements in ERG amplitudes and retinal thickness, regarding the discussion, it would be valuable to briefly address any potential limitations, such as the short duration of follow-up or the use of only male db/db mice, which might affect the generalizability of the findings, finally, a careful proofreading is recommended to correct minor typographical inconsistencies (for example, spacing in terms like “hiPSC- ECFCs” and occasional formatting artifacts in section numbers), these adjustments are minor and do not detract from the scientific value or originality of the work, and I recommend acceptance after minor revision
Author Response
Reviewer 2
I suggest a few minor revisions that may enhance the clarity and impact of the work, for instance, it would be helpful to ensure consistent use and definition of abbreviations such as hiPSC, CD34⁺, ECFCs, KNA⁺, and DR at their first mention in the abstract and throughout the main text to facilitate understanding,
Response: We have carefully edited the manuscript to ensure that each abbreviation is defined and that there is consistency for that abbreviation throughout the text.
in the methods section, consider clarifying the rationale for the chosen dosages and combination ratios of injected cell types, since this may interest readers wishing to replicate or compare protocols,
Response: The concentration of cells was based on previous studies performed using freshly isolated peripheral blood derived CD34+ cells and culture derived ECFCs[17]. (lines 120-121).
additionally, although the proteomic analysis via RPPA is described in detail, a brief explanation of how pathway enrichment and clustering were performed and interpreted would make this section more accessible to non-specialist readers.
Response: We appreciate this reviewer’s comment and have rewritten the section 3.4 Protein Expression Profiling Reveals Interactional Effects of Combination CD34+ and ECFC Therapy (lines 287-325) to focus on presenting the approach and results in a clear manner with its relevance to the conclusion of the manuscript.
for the results, the text could benefit from more explicit cross-referencing to the corresponding figures and tables, especially when discussing quantitative improvements in ERG amplitudes and retinal thickness.
Response: Each figure panel of Figure 3 is mentioned in the results text.
regarding the discussion, it would be valuable to briefly address any potential limitations, such as the short duration of follow-up or the use of only male db/db mice, which might affect the generalizability of the findings,
Response: A limitation paragraph has been added lines 414-418.
finally, a careful proofreading is recommended to correct minor typographical inconsistencies (for example, spacing in terms like “hiPSC- ECFCs” and occasional formatting artifacts in section numbers), these adjustments are minor and do not detract from the scientific value or originality of the work, and I recommend acceptance after minor revision
Response: We appreciate this reviewer’s comment and have carefully edited this revision.
Reviewer 3 Report
Comments and Suggestions for Authors
This manuscript reports a potentially important study that shows repair of capillaries in the retina with diabetic retinopathy (DR). A mouse model of DR is used and human pluripotent stem cells are used.
The study is well performed and well described, the fluorescence images are beautiful, the reference list contains relevant citations both from the distant past and the recent years. Therefore, I recommend publication. I have a few comments that can be easily addressed:
- Numbers of mice used in the various experimental groups are missing. I could only find in lines 242-258 numbers that likely are numbers of animals, but in M&M or elsewhere in the Results section clear numbers of animals are missing
- I find the title not very clear. "hIPSC-derived cell combination target....'' is quite mysterious. I advise to use a more understandable title.
- The abstract is also difficult to read. Too many abbreviations and a too lengthy text is used. I recommend a summary of max 200 words with only a few abbreviations
Author Response
Reviewer 3
- Numbers of mice used in the various experimental groups are missing. I could only find in lines 242-258 numbers that likely are numbers of animals, but in M&M or elsewhere in the Results section clear numbers of animals are missing.
Response: Thank you for bringing this to our attention and we have now added the number of mice/cohorts in the methods section and in the results in each of the figure legends.
- I find the title not very clear. "hiPSC-derived cell combination target....'' is quite mysterious. I advise using a more understandable title.
Response: We have changed the title to “Targeting Diabetic Retinopathy with Human iPSC-Derived Vascular Reparative Cells in a Type 2 Diabetes Model”
- The abstract is also difficult to read. Too many abbreviations and a too lengthy text is used. I recommend a summary of max 200 words with only a few abbreviations.
Response: As request, we have simplified the abstract (as much as possible) and shortened the length.
Round 2
Reviewer 1 Report
Comments and Suggestions for Authors
The authors have addressed all my comments satisfactorily. I recommend acceptance of the manuscript in its current form.